# Injectable Thermosensitive Chitosan-Collagen Hydrogel as A Delivery System for Marine Polysaccharide Fucoidan

**DOI:** 10.3390/md20060402

**Published:** 2022-06-18

**Authors:** Julia Ohmes, Lena Marie Saure, Fabian Schütt, Marie Trenkel, Andreas Seekamp, Regina Scherließ, Rainer Adelung, Sabine Fuchs

**Affiliations:** 1Experimental Trauma Surgery, Department of Orthopedics and Trauma Surgery, University Medical Center Schleswig-Holstein, 24105 Kiel, Germany; julia.ohmes@uksh.de (J.O.); andreas.seekamp@uksh.de (A.S.); 2Functional Nanomaterials, Institute for Materials Science, Kiel University, Kaiser Str. 2, 24143 Kiel, Germany; lms@tf.uni-kiel.de (L.M.S.); fas@tf.uni-kiel.de (F.S.); ra@tf.uni-kiel.de (R.A.); 3Department of Pharmaceutics and Biopharmaceutics, Kiel University, 24118 Kiel, Germany; mtrenkel@pharmazie.uni-kiel.de (M.T.); rscherliess@pharmazie.uni-kiel.de (R.S.)

**Keywords:** fucoidan, chitosan, hydrogel, β-GP, collagen, thermosensitive, injectable, regenerative medicine, MSC

## Abstract

Fucoidans, sulfated polysaccharides from brown algae, possess multiple bioactivities in regard to osteogenesis, angiogenesis, and inflammation, all representing key molecular processes for successful bone regeneration. To utilize fucoidans in regenerative medicine, a delivery system is needed which temporarily immobilizes the polysaccharide at the injured site. Hydrogels have become increasingly interesting biomaterials for the support of bone regeneration. Their structural resemblance with the extracellular matrix, their flexible shape, and capacity to deliver bioactive compounds or stem cells into the affected tissue make them promising materials for the support of healing processes. Especially injectable hydrogels stand out due to their minimal invasive application. In the current study, we developed an injectable thermosensitive hydrogel for the delivery of fucoidan based on chitosan, collagen, and β-glycerophosphate (β-GP). Physicochemical parameters such as gelation time, gelation temperature, swelling capacity, pH, and internal microstructure were studied. Further, human bone-derived mesenchymal stem cells (MSC) and human outgrowth endothelial cells (OEC) were cultured on top (2D) or inside the hydrogels (3D) to assess the biocompatibility. We found that the sol-gel transition occurred after approximately 1 min at 37 °C. Fucoidan integration into the hydrogel had no or only a minor impact on the mentioned physicochemical parameters compared to hydrogels which did not contain fucoidan. Release assays showed that 60% and 80% of the fucoidan was released from the hydrogel after two and six days, respectively. The hydrogel was biocompatible with MSC and OEC with a limitation for OEC encapsulation. This study demonstrates the potential of thermosensitive chitosan-collagen hydrogels as a delivery system for fucoidan and MSC for the use in regenerative medicine.

## 1. Introduction

Bone tissue has good self-healing capacities [1]. However, large and irregular fractures or diseases such as diabetes can affect the regeneration of the tissue. It is estimated that 5–10% of bone fractures fail to heal, causing long-term disability in the patient [2,3]. The local application of bioactive compounds or growth factors can stimulate molecular processes to support tissue regeneration [4,5]. Fucoidans, sulfate-rich polysaccharides from the cell wall of brown algae, have shown to be promising candidates for the use in regenerative medicine [6,7]. These polysaccharides exhibit multifaceted bioactivities in processes which are essential for successful bone regeneration, including angiogenesis, osteogenesis, and inflammation [8,9,10,11]. For the local application of fucoidan, an appropriate delivery system is needed which temporarily immobilizes and releases the molecule at the site of interest. Hydrogels serve this purpose well and are extensively studied for use in regenerative medicine for various reasons. Similar to the extracellular matrix (ECM), these porous three-dimensional networks are able to store large amounts of water and solutes, allow the exchange of substances, as well as the ingrowth of cells from adjacent tissues. Due to their flexibility, hydrogels can adapt to the shape of bone defects and support the regeneration by delivering bioactive compounds or stem cells [12,13].

Hydrogels can be divided into physically and covalently cross-linked networks. Even though often mechanically less stable, physically cross-linked hydrogels do not require the use of harsh chemicals and the molecular interactions are reversible, resulting in more biocompatible and easier degradable materials [14]. Naturally occurring molecules as building elements for hydrogels often offer advantages over synthetic components. They are characterized by an adequate biocompatibility, degradation, and structural resemblance with physiological structures such as bone [15]. A naturally occurring polymer which has the ability to form physically cross-linked porous networks is chitosan. Further, chitosan has anti-microbial properties and its cationic charge allows the interaction with anionic molecules, such as growth factors, receptors, and components from the ECM [16,17,18]. Due to these reasons, chitosan represents an interesting material for hydrogels.

Especially injectable hydrogels have received attention amongst researches in the last years. These hydrogels are liquid under specific circumstances, allowing the application via a minimally invasive injection. The gelation of such hydrogels occurs upon a specific stimulus, such as radiation, pH, or temperature change [19,20,21]. In contrast to solid implant materials, the injectable hydrogel can adapt the exact and unique shape of individual bone defects and complex implantation surgeries are avoided [22]. Together with anionic β-glycerophosphate (β-GP), chitosan forms an ionically cross-linked thermosensitive hydrogel. Hence, the hydrogel system is liquid and injectable at low temperatures, but gelates upon temperature increase [23]. Depending on the exact hydrogel composition, the gelation temperature mostly varies between 20 and 45 °C [24]. During sol-gel transition, electrostatic interactions between chitosan and β-GP and hydrogen bonds between chitosan chains are established. Modulations of hydrophobic interactions due to the glycerol moieties are responsible for the thermosensitivity [25,26,27].

Chitosan alone does not favor strong cell adhesion and spreading. One way to enhance these properties is based on the fabrication of composite materials that benefit from the combined advantages of the individual substances. The integration of collagen type I into the chitosan-β-GP system increases the biocompatibility, cell adhesion, as well as gelation properties of the hydrogel [28,29]. Collagen type I is the main structural component of the ECM and plays a major role in defining the mechanical properties of the bone and its fracture susceptibility [30]. Different studies have pointed out the potential of thermosensitive chitosan–collagen hydrogels for the use in bone regeneration and tissue engineering [28,31].

In the current study, we developed a hydrogel-based delivery system for fucoidan which is injectable at low temperatures and starts to gelate after 1 min at 37 °C. We found that physicochemical properties of the hydrogels were not affected by the integration of fucoidan. Only very high fucoidan concentrations of 500 µg/mL decreased the swelling capacity and increased the turbidity of the hydrogel. Approximately 60% and 80% of the fucoidan was released from the hydrogel after two and six days, respectively. The biocompatibility of the hydrogels was evaluated using human outgrowth endothelial cells (OEC) isolated from peripheral blood. Endothelial cells are essential for bone regeneration as they build the vasculature which transports nutrients, oxygen, and minerals into the affected tissue [32]. Further, the biocompatibility of the hydrogels was tested using human bone-derived mesenchymal stem cells (MSC) which were differentiated towards the osteoblast lineage. MSC represent another important cell type during bone healing as they regulate the angiogenic activity of OEC in a paracrine manner and serve as precursors for bone cell differentiation [33]. We found that OEC seeded on top of the hydrogel adhered and were vital while encapsulated OEC were characterized by a round and small shape. MSC encapsulated into or seeded on top of the hydrogel adhered, spread, and were vital after six days. These results indicate that the presented hydrogel could be used as a delivery system for fucoidan and stem cells for bone tissue engineering.

## 2. Results

### 2.1. Preparation and Physicochemical Characterization of the Fucoidan Delivery System

In the presented study, an injectable delivery system for fucoidan and stem cells based on chitosan, collagen, and β-GP was developed. To avoid fucoidan aggregates, it was necessary to premix the fucoidan solution with the β-GP solution immediately before adding it to the chitosan–collagen mixture. Preliminary experiments revealed that a β-GP content of 7% resulted in adequate gelation times. As demonstrated in Figure 1a, the clear sol turned into an opaque hydrogel at 37 °C. The tube-inverting method was applied to roughly estimate the time until gelation. After 3–4 min, flow of the sols could not be observed any longer, indicating the gelation of the hydrogels. The incorporation of fucoidan had no visible influence on the flow characteristics (Figure 1b). To study the gelation time and duration of the gelation process in more detail, time sweep tests were performed at 37 °C. Therefore, the sol was pipetted onto the preheated platform and the storage (G′) and loss modulus (G″) were measured for 16 min. The gel point is reached when the storage modulus, describing the elastic portion of the material, becomes larger than the loss modulus, describing the viscous portion of the material. As shown in Figure 1c, the sol-gel transition point is marked with a dashed line. Over time, storage and loss modulus increased, indicating the hydrogel formation, until reaching a plateau. The end of the gelation process is marked with another dashed line. It was found that the gel point of hydrogels without fucoidan occurred after approximately 75 s at 37 °C. The gel point of hydrogels containing 100 µg/mL fucoidan occurred after approximately 59 s at 37 °C (Figure 1d). The gelation process of both hydrogels, with and without fucoidan, lasted approximately for 6 min until reaching a stable state (Figure 1e).

Further rheological measurements were performed to determine the exact gelation temperature of the hydrogels without and with fucoidan. Temperature sweep tests as shown in Figure 2a revealed that hydrogels without fucoidan started to gelate at 36.8 °C. Hydrogels containing 100 µg/mL fucoidan already started to gelate at 35.3 °C. These results indicate that the incorporation of fucoidan into the hydrogel tentatively decreased the temperature at which the gel point is reached (Figure 2b).

Next, the pH, swelling properties, and the turbidity of the hydrogels containing different concentrations of fucoidan were examined. As shown in Figure 3a, the pH of the sols was approximately 6.8. After gelation, the pH (~6.6) was slightly more acidic. The incorporation of fucoidan had no impact on the pH before and after gelation. The swelling capacities of the hydrogels were quantified by determining the equilibrium swelling ratio (ESR). Therefore, hydrogels were lyophilized and rehydrated in PBS until reaching equilibrium. The ESR is defined as ESR = (w_s_ − w_d_/w_d_) with w_s_ and w_d_ being the weight of the swollen and the dried gels, respectively. High fucoidan concentrations affected the swelling capacities of the hydrogel as demonstrated in Figure 3b. Hydrogels containing 500 µg/mL fucoidan had three times lower ESR than hydrogels without fucoidan. Incorporation of high-fucoidan concentrations also affected the appearance of the hydrogel as depicted in Figure 3d. Hydrogels containing 500 µg/mL fucoidan were visibly more turbid. Absorption measurements revealed that hydrogels with 500 µg/mL fucoidan were four times more turbid than hydrogels with 0, 10, or 100 µg/mL fucoidan (Figure 3c).

The internal microstructure of the hydrogels was studied by scanning electron microscopy. Therefore, samples were dried using critical point drying beforehand. The incorporation of fucoidan caused no visible changes in the morphology of the hydrogels. Figure 4 shows the morphology of a hydrogel without and with 100 µg/mL fucoidan. All hydrogels consisted of fluffy spheres that were entangled by thin long filaments. The distribution of particles and filaments seemed to occur randomly. The particles tended to merge together into larger aggregates causing the formation of irregular pores. The fluffy spheres and thin filaments likely represent the chitosan and collagen portion, respectively.

In difference to many proteins, no simple and established procedure exists to quantify fucoidans, especially inside a hydrogel. Consequently, it is also challenging to determine the fucoidan release from a hydrogel over time. To prove the incorporation of fucoidan into the hydrogel, a protocol published by Yamazaki and colleagues [34] was adapted. Yamazaki and colleagues demonstrated that the nucleic acid stain SYBR Gold can be used to quantify the concentration of fucoidan solutions. Similar to DNA, the fluorescent dye binds to the fucoidan molecules. Subsequently, the amount of bound dye can be quantified by measuring the fluorescence. The detected fluorescence has a linear relation to the amount of fucoidan for a specific range of concentrations and can therefore be interpolated using a standard curve with known fucoidan concentrations. We were able to adapt the method to detect fucoidan inside the hydrogel. Even though, the determination of absolute fucoidan concentrations inside the hydrogel using a standard curve was not possible, Figure 5a clearly shows that hydrogels with more incorporated fucoidan resulted in an increased fluorescent signal. The same method was applied to study the amount of fucoidan inside the hydrogel over a time period of six days. The results are shown in Figure 5b. On day two, approximately 40% of the initially detected fucoidan were still incorporated inside the gel. On day six, only 20% of the initially measured fucoidan could be detected. These results indicate that fucoidan was released from the hydrogel over time.

### 2.2. Biocompatibility of the Fucoidan Delivery System with Primary Bone-Derived MSC and Human Endothelial Cells

To assess the biological compatibility of the chitosan–collagen hydrogel containing different concentrations of fucoidan, human bone-derived MSC and human endothelial cells were seeded on top of the hydrogels (2D culture) or incorporated into the hydrogels (3D culture) and observed using life/dead staining. The interaction of MSC with the hydrogel was additionally studied using scanning electron microcopy.

Life/dead staining of 2D-cultured MSC on day two and six revealed that cells adhered to the hydrogels as shown in Figure 6a. Incorporated fucoidan had no visible effect on the cell adhesion and viability. On day six, MSC spreading and adhesion was increased compared to day two. In a next step, the viability of MSC encapsulated into the hydrogel was studied on day two and six. As shown in Figure 6b, MSC reacted more sensitive to an encapsulation into the hydrogel than to the 2D culture. The red stained cells on day two and six indicate that dead cells remained trapped inside the hydrogel. Vital MSC adhered and elongated well inside the hydrogels. Similarly to the 2D culture, cell adhesion and spreading of the vital cells were increased from day two until day six. Again, fucoidan incorporation had no visible effect on the biocompatibility of the hydrogels. Scanning electron microscopy was applied to study the interaction of MSC with the hydrogel on a microscopic scale. As visualized in Figure 6c, MSC were completely embedded into the hydrogel material. Cellular interaction dominantly occurred with the thin filaments which most likely represent the collagen portion of the hydrogel.

2D-cultured OEC were vital and adhered to the hydrogels as shown in Figure 7a. The amount of adherent cells on day two however was larger than on day six, indicating OEC detachment over time. Similar to the culture of MSC, integration of fucoidan into the hydrogels did not change the biocompatibility. OEC were as vital on hydrogels without fucoidan as on hydrogels which contained fucoidan. As visualized in Figure 7b, 3D-cultured OEC were mostly characterized by a round and small shape which did not change from day two until day six. Different to the MSC, OEC were not able to develop an adherent and elongated phenotype inside the hydrogels. Integration of fucoidan into the hydrogels had no effect on the encapsulated OEC.

## 3. Discussion

The multifaceted biological activities of fucoidan on, for example, angiogenesis [35,36] and osteogenesis [11,37] are reported in many studies. To benefit from fucoidan’s bioactivities, a delivery system is needed which concentrates and immobilizes the molecule inside the tissue of interest. In this context, we developed a hydrogel-based delivery system for fucoidan and studied its suitability for use in regenerative medicine. The composite hydrogel consisted of chitosan and collagen type I; β-GP was added into the chitosan-collagen mixture to achieve thermosensitivity [23]. The effect of fucoidan incorporation on different physicochemical characteristics such as gelation time, gelation temperature, pH, appearance, swelling, and internal microscopic structure were studied. To evaluate the biocompatibility of the hydrogels, human bone-derived MSC and OEC isolated from peripheral blood were cultured on top and inside the hydrogels for six days. Fluorescence microscopy and SEM were applied to study the viability of the cells and their interaction with the hydrogel material.

Preliminary experiments have demonstrated that increased β-GP concentrations reduced the time until gelation. Accordingly, many studies have shown that β-GP has an impact on the gelation time [38,39,40]. It is however also reported that very high β-GP contents can be toxic to cells. Yan and colleagues demonstrated an impaired viability of rat BMSC on hydrogels containing 80% β-GP [41]. Ahmadi et al. found that hydrogels containing >10% β-GP resulted in increased cytotoxicity for goat MSC [38]. Our results showed that an amount of 7% β-GP resulted in an adequate gelation time and was not cytotoxic to our model cells.

Rheological measurements revealed that hydrogels containing 100 µg/mL fucoidan gelated after 1 min at 37 °C. This was comparable to hydrogels without fucoidan which needed around 75 s to reach the gel point. This gelation time seems appropriate for medical applications; it is long enough to allow the application via injection and to allow the gel adapting to the defect shape. On the other hand, it is short enough to prevent long waiting times and to avoid the immediate diffusion of integrated fucoidan.

Temperature sweep tests demonstrated that the gelation temperature for hydrogels containing 100 µg/mL fucoidan was 35.3 °C. The gel point of hydrogels without fucoidan occurred at 36.8 °C. Hence, hydrogels with fucoidan tentatively started to gelate at slightly lower temperatures than gels without fucoidan. The gelation of the hydrogels at 37 °C or slightly lower temperatures is advantageous for a medical application. The cooled sol can be applied at room temperature, but it will quickly start to gelate once exposed to the physiological body temperature. Several studies have shown that the gelation temperature is determined by a combination of different parameters, such as β-GP content, chitosan concentration, and molecular weight, as well as degree of deacetylation [39,42,43]. Increased β-GP content and increased chitosan concentrations result in a decreased gelation temperature [39,44]. Further, higher molecular weight and degree of deacetylation of chitosan decrease the gelation temperature as well [42].

Even though, aqueous fucoidan solutions exhibit low viscosity [43], fucoidan alone is not able to form hydrogels. Due to its sulfate residues, fucoidan usually has an overall negative charge. Together with oppositely charged polysaccharides such as chitosan, hydrogels or films can be formed based on electrostatic interactions [43,45]. The results from the rheological measurements, swelling experiments, and the visual appearance (turbidity) indicate that the integration of fucoidan into the chitosan hydrogel system increased the degree of interconnectivity inside the gel. Swelling experiments revealed that hydrogels containing high amounts (500 µg/mL) of fucoidan absorbed less water than hydrogels with no or less fucoidan. We assume that the additional molecular interactions resulted in faster gelation at lower temperatures and that the increased matrix interconnectivity hindered water uptake into the gel, therefore reducing the swelling capacities of hydrogels containing 500 µg/mL fucoidan.

To our knowledge, no study exists which developed and characterized thermosensitive chitosan–collagen hydrogels with integrated fucoidan. To date, most studies investigated the potential of fucoidan as a component of nanoparticles for the delivery of specific drugs. The putative applications are manifold: anti-cancer, anti-bacterial, to treat pulmonary diseases or diabetes [46,47,48,49]. However, some studies developed hydrogel systems containing fucoidan. Carvalho and colleagues prepared ionically-crosslinked hydrogels with collagen, chitosan and fucoidan and found that gels containing all three polysaccharides possessed better mechanical properties than hydrogels with only two polysaccharides [45]. Sezer and colleagues produced chitosan–fucoidan hydrogels for the treatment of dermal burns [50]. In contrast to our results, Sezer et al. found that hydrogels with fucoidan absorbed more water, indicating greater swelling capacities. This was associated with the hydrophilic nature of fucoidan. Another explanation for the different result in the experiments of Sezer and colleagues is the absence of collagen and β-GP which are additional crosslinking molecules. Other studies describe the preparation of chitosan–fucoidan hydrogels for the encapsulation of angiogenic growth factors [51] or the development of alginate–fucoidan composite hydrogels for enhancing chondrogenesis of stem cells [52]. Additionally, some approaches are described which utilize covalent cross-linking to create hydrogels containing fucoidan. Lu and colleagues prepared genipin crosslinked hyaluronic acid–fucoidan–gelatin hydrogels for the delivery of platelet-rich plasma [53]. A study from Hsu and colleagues presented the production of methacrylated hyaluronan and fucoidan to create photo-crosslinkable hydrogels [6].

In almost all biomaterial approaches which include fucoidan, the polysaccharide rather represents a structural element with beneficial properties, such as enhancing the bioactivity of the encapsulated drug or altering the mechanical properties of the hydrogel [45,51]. To our knowledge, no study exists which integrates fucoidan into a delivery system and examines its release comparable to a drug. Our experiments indicated that approximately 60% of the fucoidan was released from the hydrogel within two days, and 80% of the fucoidan was released after six days. These results suggest that fucoidan was not permanently immobilized inside the hydrogel. Its interactions with the other components were reversible, allowing the release of fucoidan into the supernatant over a specific period of time.

To use a biomaterial in the medical context, biocompatibility is a crucial requirement. It is defined as “the ability of a material to perform with an appropriate host response in a specific application” [54]. Interaction of human cells with the material is especially important for applications dealing with tissue regeneration. Here, the biomaterial is continuously in direct contact with the injured tissue, providing a platform for adjacent cells to migrate into the defect and initiate its regeneration [13]. Life/dead stainings and SEM revealed that human MSC were vital when cultured inside (3D) or on top (2D) of the injectable chitosan–collagen hydrogels with integrated fucoidan. Further, we showed by fluorescence microscopy that MSC adhered and had an elongated phenotype, indicating a good biocompatibility of the material with MSC. Life/dead stainings showed that 2D-cultured OEC were vital and adhered to the material. However, encapsulated OEC had a round and small phenotype indicating difficulties of OEC to spread well inside the material. These results demonstrate that MSC can be encapsulated along with fucoidan into the hydrogel to support healing processes. However, to encapsulate OEC alone, structural adjustments of the hydrogel are needed. In addition, a co-encpasulation of OEC with another cell type such as MSC might help to improve the biocompatibility.

## 4. Conclusions

To conclude, we were able to develop an injectable hydrogel-based delivery system for fucoidan. The hydrogel was injectable at room temperature and gelated within 1 min at 37 °C. The incorporation of fucoidan into the hydrogel had only minor impacts on the determined physicochemical properties. Only the incorporation of high fucoidan concentrations (500 µg/mL) increased the turbidity and decreased the swelling capacity of the hydrogel. Further, the biomaterial was compatible with human MSC and OEC with a limitation for OEC encapsulation. Due to its short gelation time, physiological gelation temperature, and biocompatibility, the hydrogel represents a promising delivery system for fucoidan for the use in regenerative medicine.

## 5. Materials and Methods

### 5.1. Ethical Approval

The use of human material was approved by the local ethics committee of the University Medical Center Schleswig-Holstein. Isolation of primary cells from human tissue was performed with the consent of the donors.

### 5.2. Hydrogel Preparation

All steps were performed on ice and under constant stirring. Two percent (in 0.1 M acetic acid) 4 °C cooled chitosan (Chitosan 95/100, Heppe Medical Chitosan GmbH, Halle, Germany) solution was pipetted into a glass vial, followed by addition of rat tail collagen type I (Corning, Bedford, MA, USA) to reach a final concentration of 1.5 mg/mL. Cold 50% β-glycerophosphate (Sigma-Aldrich, St. Louis, MO, USA, end concentration 7%) was mixed with aqueous 5 mg/mL fucoidan solution (≥95% from *Fucus vesiculosus*, Sigma-Aldrich, end concentrations 0–500 µg/mL) and immediately added dropwise into the chitosan–collagen mixture. The mixture was incubated on ice under constant stirring for 15 min to ensure proper mixing of all components. The sol was always used on the same day for experiments.

### 5.3. Characterization of the Hydrogel

#### 5.3.1. Gelation Time by Tube-Inverting Method

The gelation time was roughly estimated by using the tube-inverting method. The sol was incubated in a water bath at 37 °C and inverted every minute. The sol was defined as gelated when no flow could be observed any longer.

#### 5.3.2. Rheological Measurements

The rheological studies were performed with the CVO 120 HRNF (Bohlin Instruments, Pforzheim, Germany) plate-to-plate rheometer (plate diameter: 40 mm). One mL of sample covered with a solvent trap to prevent evaporation was used for each measurement. The storage (G′) and loss moduli (G″) were determined to characterize the viscoelastic properties of the hydrogels with and without fucoidan. At the beginning, an amplitude sweep test was carried out at 37 °C and at a constant frequency of 1 Hz to determine the viscoelastic region of the hydrogels. The start of the gelation process and its duration were assessed by time sweep tests at 37 °C, 1 Hz, and 5% strain. To determine the exact gelation temperature of the hydrogels, temperature sweep tests were performed. Therefore, G′ and G″ were measured at 10–60 °C (2 °C/min) at a constant frequency of 1 Hz and 5% strain.

#### 5.3.3. pH

The pH of the sols was measured before and after gelation using a SevenEasy pH meter S20 with an InLab Micro Pro-ISM pH electrode (Mettler-Toledo GmbH, Gießen, Germany).

#### 5.3.4. Swelling

To quantify the swelling capacity of the hydrogels, the equilibrium swelling ratio (ESR) was determined. Therefore, hydrogels were prepared and quick-frozen in liquid nitrogen, followed by lyophilization (ScanVac CoolSafe Freeze Dryer, LaboGene, Allerød, Denmark). Dried hydrogels were weighted and incubated in PBS at 37 °C for three days. After three days, the weight of the swollen hydrogels was determined. The ESR was calculated using the following equation: ESR = ((w_s_ − w_d_)/w_d_), with w_s_ and w_d_ = weight of swollen and dried hydrogel, respectively.

#### 5.3.5. Turbidity

The turbidity of the sol was quantified by measuring the absorbance at 600 nm using a plate reader (infinite M200 Pro, Tecan, Männedorf, Switzerland).

#### 5.3.6. SEM

Hydrogels were prepared in 8-well slides and incubated in PBS for 24 h. On the next day, hydrogels were fixed for 30 min at room temperature using 3% glutaraldehyde (Sigma-Aldrich) in 4% paraformaldehyde (Morphisto, Frankfurt am Main, Germany). After 3 times washing for 5 min with PBS, aqueous buffer was replaced gradually by ethanol (Merck, Darmstadt, Germany). Each ethanol dilution (50%, 60%, 70%, 80%, 90%, 95%) was incubated on the hydrogels for 15 min until reaching 100% ethanol. Subsequently, hydrogels were carefully removed from the wells and cut in half for imaging the gel morphology at the cross-section or left intact for imaging the surface of the gels. Hydrogels were dried using a critical-point dryer (Leica EM CPD300, Nussloch, Germany) and glued to sample holders using conductive carbon tape. Subsequently, samples were sputtered with gold for 90 s and imaged with a Supra 55VP (Zeiss, Jena, Germany).

### 5.4. Fucoidan Detection Inside the Hydrogel

Fucoidan incorporation into the hydrogel was proven using the SYBR Gold Nucleic Acid Gel Stain (Invitrogen, Eugene, Oregon, USA) using an adapted protocol from Yamazaki and colleagues [34]. One-hundred µL sol was pipetted into a 96-well plate and gelated at 37 °C for 30 min. SYBR Gold stain solution was diluted 1:625 in 80 mM Tris-HCl (pH 7.5) and 50 µL was applied onto the hydrogels. After 1 h incubation at room temperature, the staining solution was removed and hydrogels were washed with PBS. The fluorescence was measured with a plate reader at 470 nm and 600 nm excitation and emission wavelength, respectively.

For the fucoidan release assay, 100 µL sol was pipetted into a 96-well plate and gelated at 37 °C. Sols were incubated at 37 °C in PBS. On days 0, 2, and 6, fucoidan was detected using SYBR Gold Nucleic Acid Gel Stain as described above.

### 5.5. Primary Cell Isolation and Cultivation

#### 5.5.1. Human Mesenchymal Stem Cells

Human mesenchymal stem cells (MSC) were isolated from cancellous bone as described before [33]. MSC were cultured in collagen type I-coated flasks using Dulbecco’s Modified Eagle’s Medium (DMEM)/Ham’s F-12 (PAN Biotech, Aidenbach, Germany) containing 20% FBS (Sigma-Aldrich, Steinheim, Germany) and 1% Penicillin/Streptomycin (gibco, Grand Island, NY, USA). The FBS content was reduced to 10% after the first sub-culture. After the second sub-culture, MSC were differentiated into an osteoblast-like lineage using osteogenic differentiation medium (ODM: DMEM/Ham’s F-12 with 10% FBS, 1% Penicillin/Streptomycin, 50 µM L-ascorbic acid 2-phosphate (Sigma-Aldrich), 10 mM β-GP (Sigma-Aldrich), and 0.1 µM dexamethasone (Sigma-Aldrich). The medium was exchanged every second or third day.

#### 5.5.2. Human Outgrowth Endothelial Cells

Human outgrowth endothelial cells (OEC) were isolated from peripheral blood as described before [55]. OEC were cultured in fibronectin-coated flasks using endothelial growth medium-2 (EGM-2), containing endothelial basal medium (EBM-2), EGM-2-associated supplements (Promocell, Heidelberg, Germany), 5% FBS, and 1% Penicillin/Streptomycin. OEC were sub-cultured every two or three days after reaching confluency. The medium was exchanged every second or third day.

### 5.6. Cell Culture Inside and on Hydrogels

#### 5.6.1. 2D Cell Culture

Sols were pipetted into the wells of an adequate dish and gelated at 37 °C for 30 min. 80,000 cells/cm^2^ were seeded on top of the hydrogel after gelation. The medium was gently exchanged each or every second day.

#### 5.6.2. 3D Cell Culture

For cell encapsulation into the hydrogel, the cell suspension was centrifuged at 400× *g* for 5 min and the pellet was resuspended in cold sol in a concentration of 10^6^ cells/mL by pipetting up and down. Sols including cells were pipetted into wells and gelated for 30 min at 37 °C. Subsequently, warm culture medium was added on top of the hydrogels. The medium was gently exchanged each or every second day.

### 5.7. Cell Life/Dead Staining

For life/dead stainings, cells were cultured 2D or 3D in µ-Slides Angiogenesis (Ibidi, Gräfelfind, Germany). On day two or six, the culture medium was gently removed and replaced by fresh culture medium containing 10 µg/mL CalceinAM (Sigma-Aldrich) and 2 µg/mL propidium iodide (Sigma-Aldrich) for staining vital and dead cells, respectively. The staining solution was incubated for 30 min at 37 °C. Subsequently, the staining solution was removed and replaced by fresh medium. Cells were imaged with the Evos FL Auto 2 fluorescence microscope (Thermo Fisher Scientific, Waltham, MA, USA).

### 5.8. Statistical Analysis

Unpaired *t*-test was used to calculate the statistical significances in Prism 7.03. (GraphPad, San Diego, CA, USA) Values were considered as significant when *p* < 0.05.

## Figures and Tables

**Figure 1 marinedrugs-20-00402-f001:**
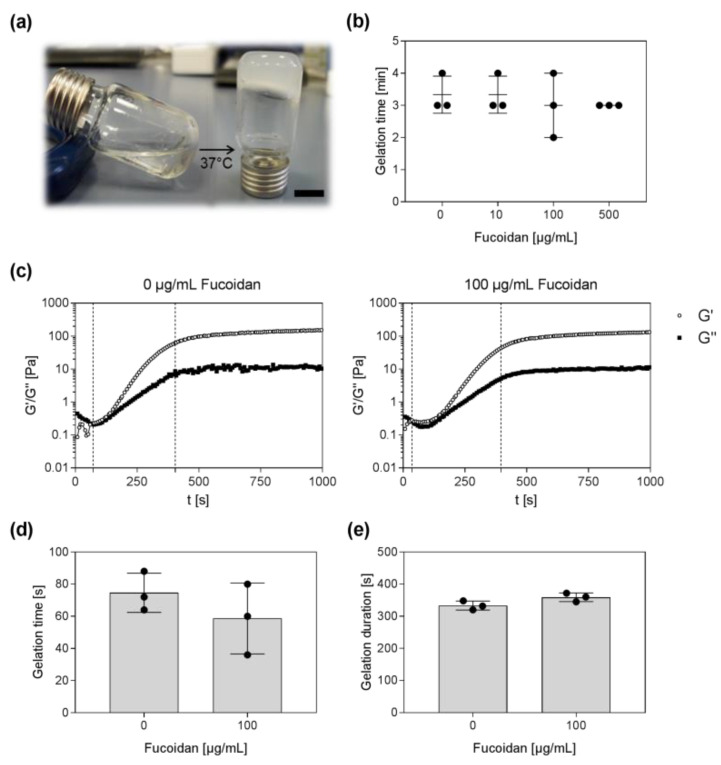
Appearance and gelation time of injectable chitosan–collagen hydrogel containing different fucoidan concentrations. (**a**) The clear liquid sol turns into an opaque hydrogel at 37 °C. Scale bar = 1 cm. (**b**) The gelation time was approximated using the tube-inverting method. The gelation time was reached when no flow was observed any longer. (**c**) Time sweep tests of hydrogels without and with 100 µg/mL fucoidan at 37 °C. The gel point (G′ (dots) > G″ (squares)) and the end of the gelation process are marked with a dashed line. (**d**) Start time of the gelation process and (**e**) gelation duration were plotted for hydrogels with and without fucoidan. Experiments were repeated three times (*n* = 3).

**Figure 2 marinedrugs-20-00402-f002:**
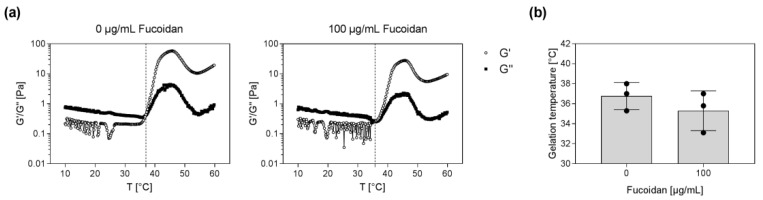
Gelation temperature of injectable chitosan–collagen hydrogel containing different fucoidan concentrations. (**a**) Temperature sweep tests were performed to determine the exact gelation temperature. The gel point (G′ (dots) > G″ (squares)) is marked with a dashed line. (**b**) The gelation temperature was plotted for hydrogels without and with 100 µg/mL fucoidan. Experiments were repeated three times (*n* = 3).

**Figure 3 marinedrugs-20-00402-f003:**
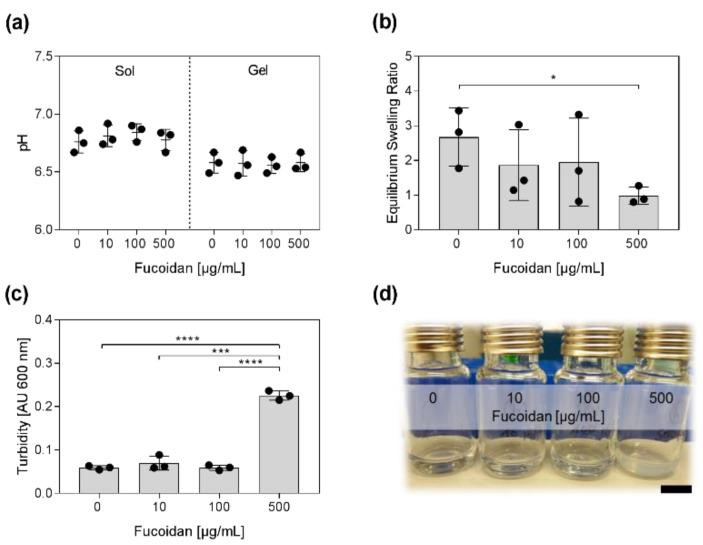
pH, swelling, and turbidity of the injectable chitosan–collagen hydrogel containing different fucoidan concentrations. (**a**) The pH was measured before and after gelation. (**b**) The swelling capacity of the hydrogels was estimated by determination of the equilibrium swelling ratio (ESR). Hydrogels were lyophilized and rehydrated until reaching equilibrium. ESR = (w_s_ − w_d_)/w_d_); w_s_, w_d_ = weight of swollen and dried hydrogel, respectively. (**c**) The turbidity of the sols was quantified by measuring the absorbance at 600 nm. (**d**) Sols containing 500 µg/mL fucoidan are visibly more turbid. Scale bar = 1 cm. Experiments were repeated three times (*n* = 3). Significances compared to the control (hydrogel with 0 µg/mL Fucoidan) were calculated with the unpaired *t*-test (* *p* < 0.05, *** *p* < 0.001, **** *p* < 0.0001).

**Figure 4 marinedrugs-20-00402-f004:**
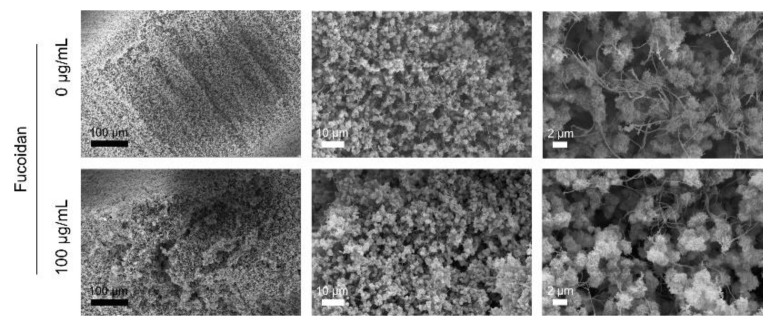
Internal microstructure of the injectable chitosan–collagen hydrogel without and with 100 µg/mL fucoidan. Hydrogels were fixed and dried using critical point drying. Samples were imaged by scanning electron microscopy. Scale bars from left to right = 100, 10, 2 µm.

**Figure 5 marinedrugs-20-00402-f005:**
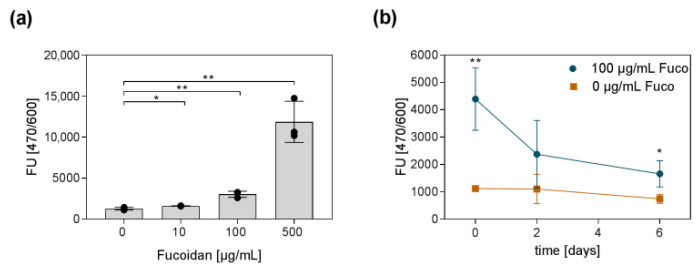
Fucoidan detection inside the hydrogel and fucoidan release from the hydrogel. The protocol from Yamazaki and colleagues [34] was adapted to detect fucoidan inside a hydrogel. (**a**) The SYBR Gold nucleic acid stain allowed detecting different fucoidan amounts inside the hydrogel. (**b**) Hydrogels with and without fucoidan were incubated in PBS for 6 days. The fucoidan content inside the hydrogel was detected on day 0, 2, and 6 using SYBR Gold. Experiments were repeated three times (*n* = 3). Significances compared to the control (hydrogel with 0 µg/mL Fucoidan) were calculated with the unpaired *t*-test (* *p* < 0.05, ** *p* < 0.01).

**Figure 6 marinedrugs-20-00402-f006:**
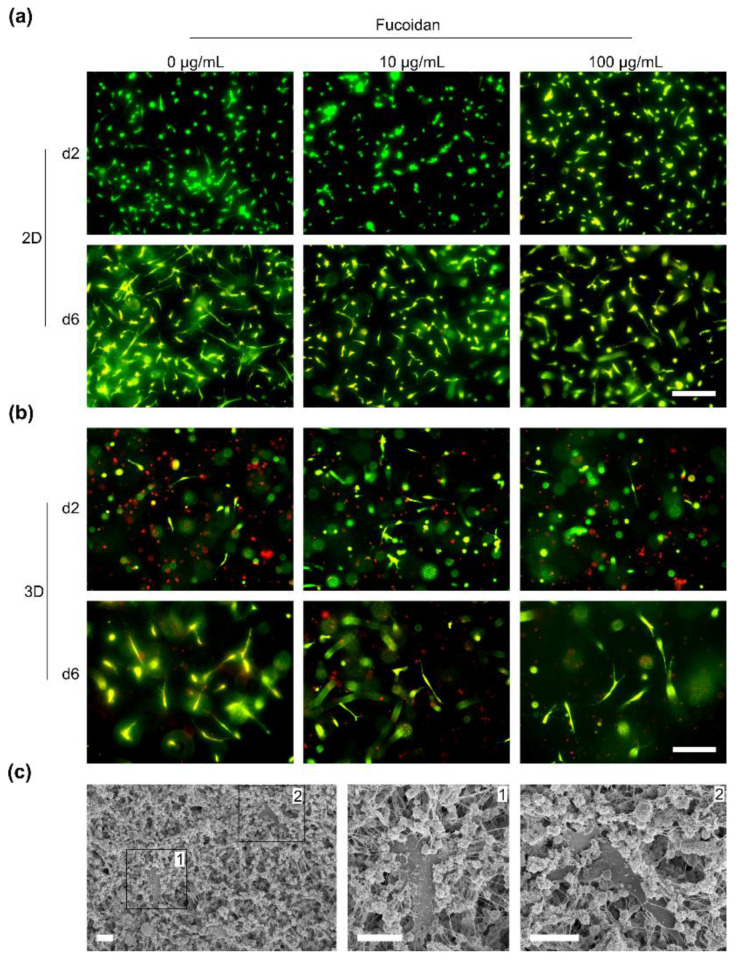
Biocompatibility of human mesenchymal stem cells (MSC) with injectable chitosan-collagen hydrogels containing different fucoidan concentrations. (**a**) MSC were cultured on top of the hydrogels (2D culture), (**b**) MSC were encapsulated into the hydrogels (3D culture). Vital (green) and dead (red) 2D- and 3D-cultured cells were visualized on day two and six using CalceinAM and propidium iodide staining, respectively. Scale bars = 200 µm. (**c**) MSC were cultivated on hydrogels containing 100 µg/mL fucoidan and visualized by scanning electron microscopy after six days. MSC in black frames numbered with 1 and 2 are shown in a higher magnification in the middle and on the left hand side. Scale bars = 10 µm.

**Figure 7 marinedrugs-20-00402-f007:**
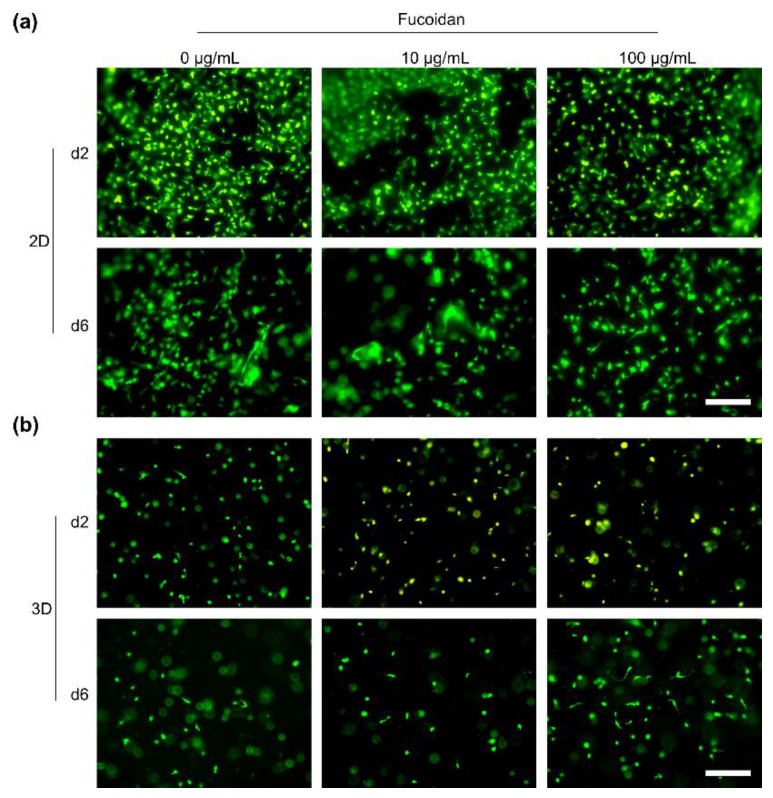
Biocompatibility of human outgrowth endothelial cells (OEC) with injectable chitosan-collagen hydrogels containing different fucoidan concentrations. (**a**) OEC were cultured on top of the hydrogels (2D culture), (**b**) OEC were encapsulated into the hydrogels (3D culture). Vital (green) and dead (red) 2D- and 3D-cultured cells were visualized on day two and six using CalceinAM and propidium iodide staining, respectively. Scale bars = 200 µm.

## Data Availability

Not applicable.

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
