# Peer review of "Injectable Thermosensitive Chitosan-Collagen Hydrogel as A Delivery System for Marine Polysaccharide Fucoidan"

_marinedrugs, 2022, doi:10.3390/md20060402_

Round 1

Reviewer 1 Report

According to the current research, fucoidan was encapsulated in a hydrogel composed of chitosan and collagen and used as a drug delivery system to induce bone regeneration. The authors claim that by encapsulating fucoidan, an excellent material that can induce bone regeneration and has high biocompatibility, it is possible to effectively induce bone regeneration and maintain it in the body using the phenomenon that chitosan-collagen becomes gel at body temperature. The experiments were well designed, and structural characterization and performance assessments were well performed. All of the main statements are supported by experimental analysis arguments. The work seems to be interesting and provide some new information for the development of fucoidan delivery for bone regeneration. It is acceptable after several minor issues are addressed.

1. In figure 6 and 7, it is better to present the survival percent of human mesenchymal stem cells (MSC).

2. In line 106, 212, there are errors of numbers of minor titles.

Reviewer 2 Report

The manuscript entitled “Injectable Thermosensitive Chitosan-Collagen Hydrogel as a Delivery System for Marine Polysaccharide Fucoidan” by Julia Ohmes and et al. devoted to the creation of fucoidan delivery system, based on chitosan, collagen and β-glycerophosphate, investigation of its physicochemical parameters and the determination of the biocompatibility of obtained injectable thermosensitive hydrogel. This work is beneficial for researchers who focus on the investigation of fucoidans and their application.

The manuscript seems well written, and contents look important.

The part of the work devoted to the preparation and physicochemical characterization of the fucoidan delivery system represented in detail. Gelation time, gelation temperature, pH, appearance, swelling and internal microscopic structure were determined and well described.

The biocompatibility of obtained injectable thermosensitive hydrogel was assessing only by cell life/dead staining of MSC and OEC cells. To confirm the perspectiveness of the delivery system for fucoidan for the use in regenerative medicine, the additional experiments on biocompatibility of injectable thermosensitive hydrogel are needed.

Reviewer 3 Report

In this manuscript is described the preparation of some injectable thermosensitive chitosan-collagen hydrogels as fucoidan delivery systems.  The objectives and motivation of this study is very well defined in the introduction. The hydrogels characterization methods are adequately described. The results are clearly presented. I recommend the paper publication after minor revision following my comments. I suggest the insertion of conclusions part as a separate section. The conclusion section should summarize the results and set them into perspective to the objectives formulated in the introduction. Moreover it has to give an outlook on the importance of the findings. 

Round 2

Reviewer 2 Report

The authors have addressed my questions and described further plans of biocompatibility experiments. The manuscript might be accepted in the present form.